# Diurnal Preference and Correlates of Multidimensional Perfectionism, Type-D Personality, and Big Five Personality Traits

Jodie C. Stevenson [1], Anna Johann [2], Asha Akram [3], Sarah Allen [4] and Umair Akram [5,*]

1 College of Social Sciences, University of Lincoln, Lincoln LN6 7DQ, UK
2 Faculty of Medicine, University of Freiburg, 79038 Freiburg, Germany
3 Faculty of Social Sciences, University of Sheffield, Sheffield S10 2TN, UK
4 Faculty of Health & Life, Northumbria University, Newcastle NE2 1UY, UK
5 College of Social Sciences & Arts, Sheffield Hallam University, Sheffield S10 2BP, UK
* Correspondence: u.akram@shu.ac.uk

**Abstract:** This study examined the extent to which the dimensions of the five-factor model, Type-D personality, and multidimensional perfectionism were associated with a diurnal preference in the general population. A sample of ($N$ = 864) individuals completed the measures of diurnal preference, multidimensional perfectionism, Type-D personality, and the Big Five traits. A correlational analysis determined that agreeableness, conscientiousness, emotional stability, organization, and personal standards were independently related to morningness. In contrast, negative affect, social inhibition, Type-D personality, and perfectionistic doubts and concerns, as well as an increased perception of critical parental evaluation, were independently related to eveningness. After accounting for the shared variance amongst the personality traits, only negative affect, conscientiousness, organization, personal standards, and parental perception were significantly associated with diurnal preference. The current outcomes offer further insight into the relationship between personality and diurnal preference. Here, we observed greater reports of adaptive personality traits in relation to morningness, whereas negative affect and perceived parental evaluation and criticism were related to eveningness. As the first study to examine the relationship between Type-D personality, multidimensional perfectionism, and diurnal preference, the current outcomes should be considered preliminary.

**Keywords:** chronotype; sleep; personality





## 1. Introduction

Diurnal preference refers to a person's preferred timing of sleep–wake behavior, ranging from morning to evening types [1]. Typically, morning types elicit greater performances in the morning hours, sleep earlier in the evening, and wake earlier in the morning. Conversely, evening types report difficulties in arising early, sleeping later in the evening, and greater performances in the late afternoon and evening hours [2,3]. Personality may theoretically influence the nature of one's diurnal preference. Indeed, various personality traits are differentially associated with a greater preference for the morning or evening [4–17]. However, the literature remains sparse, often yielding mixed outcomes, perhaps due to methodological differences concerning the sample populations and scales used.

In a small sample of Cambridge university students, Matthews [4] found that a preference for eveningness was related to greater trait-like anxiety. Diaz-Morales [5] evidenced that Spanish university students classified as morning types reported greater levels of self-control whilst displaying an overconcern with making a positive impression. In contrast, those with an evening disposition reported greater creativity and a propensity for risk-taking behavior. Moreover, whilst Hasler [6] failed to demonstrate a relationship

between negative affect (i.e., an increased experience of negative emotions and poor self-concept) and evening preference in the general population, morningness was significantly related to increased positive affect. Several studies concur that increased psychopathy and Machiavellianism, aspects of the dark-triad traits, are associated with an evening preference amongst the UK general population [7], and amongst university students from the United Kingdom [8] and Germany [9]. However, most studies focus on the five-factor model of personality [10]. Here, the relationship between morningness and conscientiousness (i.e., a tendency to be reliable, well organized, and hardworking) is perhaps the most reliable to date, consistently observed in student- and general-population samples [11–17]. In addition, greater levels of agreeableness (i.e., the tendency to act warm, friendly, and tactful), emotional stability (i.e., reduced experience of emotionally reactive behavior and negative emotions), and openness to experience (tendency to be open-minded, imaginative, and creative) have been differentially reported amongst Polish [11], Estonian [12], and Italian [17] adults, and Polish [14], German [13], and Canadian [16] students.

To the best of our knowledge, the possible relationship(s) between diurnal preference and the dimensions of multidimensional perfectionism (excessive personal standards and overly critical self-evaluation) and Type-D personality (joint experience of negative affect and social inhibition) remain unexamined. Therefore, this exploratory study aimed to determine: (i) the extent to which the dimensions of the five-factor model, Type-D personality, and multidimensional perfectionism are directionally associated with a diurnal preference in the general population; (ii) any empirically supported relationships that remained after accounting for the shared variance between the traits. Considering previous outcomes [4–17], it was hypothesized that eveningness is related to the negatively perceived aspects of personality. In contrast, we expected the preference for morningness to be associated with more positively oriented traits. The present study is the first to examine the diurnal preference in relation to Type-D personality and multidimensional perfectionism. However, given the mixed evidence concerning which dimensions of the Big Five personality traits are related to the chronotype, no a priori hypotheses are made in relation to this question.

## 2. Materials and Methods

### 2.1. Sample and Procedure

A cross-sectional online questionnaire-based study was implemented, comprising questions designed to assess the relationship between the facets of perfectionism, dysfunctional beliefs about sleep, and symptoms of anxiety and insomnia. The study protocol was approved by the Sheffield Hallam University Research Ethics Committee, and all participants provided informed consent. The survey was advertised to: (a) members of the general population through social media and online forums; (b) students at four northern UK universities, through each institution's course participation scheme. $N = 951$ began the survey, which was delivered using the Qualtrics platform (Qualtrics, Provo, Utah, United States). Only complete cases were used in the analysis due to the ethical right to withdraw from the survey at any time. The data were also examined for duplicate responses based on matching IP addresses, and none were found. As such, $N = 864$ respondents who provided complete data (mean age: 22.93 ± 9.67 years, range: 18–76 years; 78% female; 54% students) were included in the final analysis. This sample size was sufficient for a 95% confidence level, exceeding our target of 500 responses, leaving an acceptable 4.5% margin of error [18].

### 2.2. Measures

The chronotype preference was determined using the 19-item Morningness–Eveningness Questionnaire (MEQ: [19]), which asks about individuals' sleep timing and schedules (e.g., "If you got into bed at 11 PM, how tired would you be?" (0 = not at all tired; 5 = very tired), and "During the first half hour after you wake up in the morning, how do you feel?" (1 = very tired; 4 = very refreshed)). The total scores range between 16 and 86, and higher scores indicate a greater preference for morningness. In contrast, lower scores indicate

a disposition for eveningness. Total scores between 16 and 41, 42 and 58, and 59 and 86 indicate a preference for morningness, neither (intermediate), or eveningness, respectively. Items are summed to form an index of the chronotype. A good level of internal consistency was yielded (Cronbach's $\alpha$ of 0.86).

Negative affect (NA), social inhibition (SI), and the dimensional Type-D interaction (NAxSI) were assessed using the 14-item DS14 [20]. Specifically, this measure comprises two seven-item subscales to measure NA (e.g., "I often feel unhappy") and SI (e.g., "I am a closed kind of person"), with a maximum score of 28 on each scale. Each item is measured on a 5-point Likert scale: 0 = false; 1 = mostly false; 2 = neutral; 3 = mostly true; 4 = true. Higher scores on each subscale indicate greater levels of the respective trait. To analyze Type-D personality as a dimensional construct, a continuous measure of Type-D personality was computed using the arithmetic product of the SI and NA scores. This is in line with recent studies examining the dimensional Type-D construct [21–24]. An assessment of the internal consistency yielded a Cronbach's $\alpha$ of 0.90 for NA, and 0.86 for SI.

The Ten-Item Personality Inventory (TIPI) was used to assess the Big Five personality dimensions [25]. The TIPI covers the personality dimensions of extroversion, agreeableness, conscientiousness, emotional stability, and openness to new experiences using independent subscales. This instrument consists of 10 items, with a common stem of "I see myself as". Each item consists of two descriptors that represent a pole of the Big Five personality dimensions, and the inventory is rated on a 7-point scale ranging from 1 (disagree strongly) to 7 (agree strongly). Many studies have supported the reliability and validity of the measure (e.g., [25]). The score for each subscale ranges from 2 to 14, with higher scores indicating a greater presence of the specific trait. An assessment of the internal consistency yielded Cronbach's $\alpha$ values of 0.74, 0.30, 0.50, 70, and 0.30 for the extraversion, agreeableness, conscientiousness, emotional stability, and openness to new experiences subscales, respectively.

The original version of the Frost (F-MPS) Multidimensional Perfectionism Scale assessed the perfectionistic traits [26]. The four-dimensional scoring approach was taken [27]. Here, the 35 items of the original F-MPS examined four dimensions on a 5-point Likert scale. The scores for each component range were as follows: concern over mistakes and doubts (CMD): 13–65; personal standards (PS): 7–35; organization (ORG): 6–30; parental expectations and criticism (PEC): 11–65. Higher scores represent a greater tendency towards perfectionism. An internal-consistency assessment yielded Cronbach's $\alpha$ values of 0.92 for CMD, 0.82 for PS, 0.91 for ORG, and 0.88 for PEC.

*2.3. Statistical Analysis*

Correlational analyses (Pearson's bivariate) examined the relationship between each personality trait and chronotype preference. Next, a bootstrapped hierarchical linear regression analysis was used to assess the predictive value of the significant correlational associations between the chronotype and assessed personality characteristics. Bootstrapping is a robust alternative to standard parametric estimates when the assumptions around the latter may be violated [28]. Bootstrapping with 1000 bias-corrected and accelerated (BCa) resamples and 95% confidence intervals was used, as this analytic approach allows for a more robust estimation of the regression coefficients [28]. Each personality trait was entered as a separate predictor in the model. Specifically, aspects of Type-D personality were entered in Step 1, the Big Five personality traits in Step 2, and multidimensional perfection in Step 3. All data were analyzed in IBM SPSS v.24.0 (IBM Corp., Armonk, NT, USA). Significance was considered at the $p < 0.05$ level.

## 3. Results

The mean scores for the chronotype preference and each personality trait are presented in Table 1. The normality of the MEQ data was assessed using a histogram, which indicated the distribution to be skewed in favor of eveningness (see Figure 1). More specifically, 7.1%

of the sample indicated a morning preference, 51.2% an intermediate preference, and 41.8% an evening preference.

**Table 1.** Means and standard deviations (SDs) for MEQ, DS14, TIPI, and F-MPS scores.

| | Mean | SD | Ranges |
|---|---|---|---|
| MEQ: Chronotype | 43.69 | 9.57 | 21–72 |
| DS14: Negative Affect | 13.89 | 6.81 | 0–28 |
| DS14: Social Inhibition | 12.46 | 6.32 | 0–28 |
| DS14: SI x NA | 195.09 | 160.51 | 0–784 |
| TIPI: Extraversion | 4.08 | 1.53 | 1–7 |
| TIPI: Agreeableness | 4.83 | 1.13 | 1–7 |
| TIPI: Conscientiousness | 4.86 | 1.32 | 1–7 |
| TIPI: Emotional Stability | 3.85 | 1.51 | 1–7 |
| TIPI: Openness to Experiences | 4.96 | 1.09 | 1–7 |
| FMPS: Concerns and Doubts | 35.76 | 10.93 | 13–65 |
| FMPS: Organization | 21.50 | 5.08 | 6–30 |
| FMPS: Personal Standards | 24.07 | 5.78 | 8–40 |
| FMPS: Parental Expectation Criticism | 19.31 | 6.85 | 8–40 |

Note: MEQ: Morningness–Eveningness Questionnaire; DS14: Distressed Personality 14; TIPI: Ten-Item Personality Inventory; FMPS: Multidimensional Perfectionism Scale.

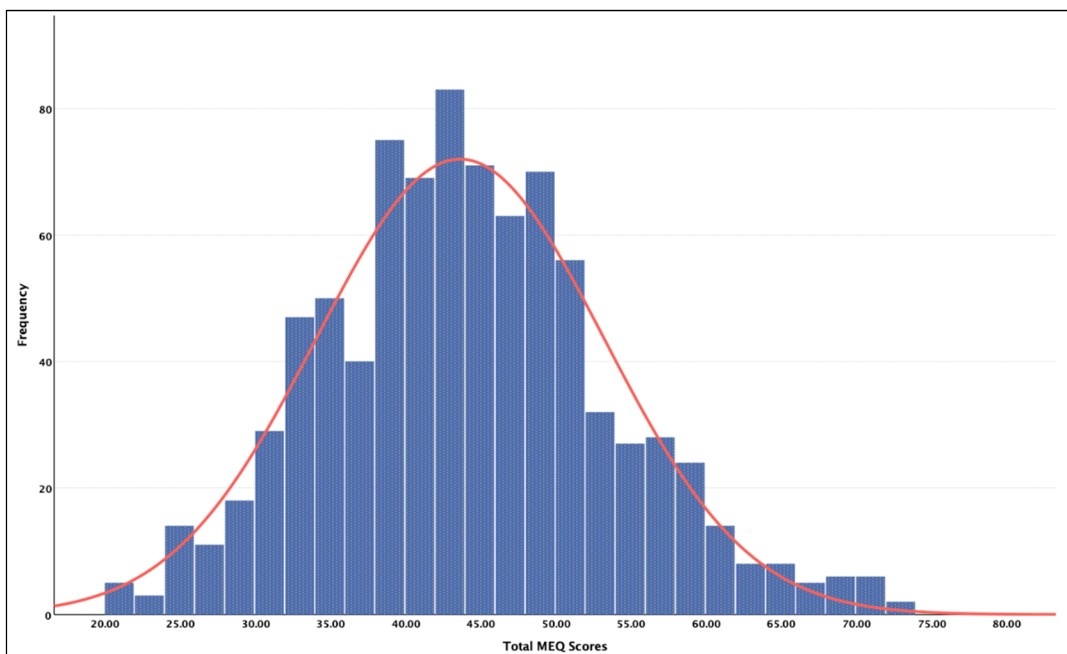

**Figure 1.** Distribution of total MEQ scores in the current sample, in which scores between 16 and 41 indicate an evening preference, scores between 42 and 58 indicate an intermediate preference, and scores between 59 and 86 indicate a morning preference.

The morning chronotype preference was negatively related to negative affect (r = −0.21, $p < 0.01$), social inhibition (r = −0.12, $p < 0.01$), Type-D interaction (r = −0.17, $p < 0.01$), perfectionistic doubts and concerns (r = −0.13, $p < 0.01$), and parental perception (r = −0.13, $p < 0.01$), and it was positively related to agreeableness (r = 0.09, $p < 0.05$), conscientiousness (r = 0.33, $p < 0.01$), emotional stability (r = 0.17, $p < 0.01$), perfectionistic organization (r = 0.28, $p < 0.01$), and personal standards (r = 0.08, $p < 0.05$). No significant relationships between the chronotype and extraversion and openness to experiences were observed (all $p$'s > 0.05), and consequently, they were not entered into the following regression analysis (see Table 2 for all other correlations).

**Table 2.** Correlations between chronotype preference and personality traits.

| | 1. | 2. | 3. | 4. | 5. | 6. | 7. | 8. | 9. | 10. | 11. | 12. |
|---|---|---|---|---|---|---|---|---|---|---|---|---|
| 1. MEQ | - | | | | | | | | | | | |
| 2. NA | −0.21 ** | - | | | | | | | | | | |
| 3. SI | −0.12 ** | 0.51 ** | - | | | | | | | | | |
| 4. SI x NA | −0.17 ** | 0.82 ** | 0.85 ** | - | | | | | | | | |
| 4. EXT | 0.02 | −0.34 ** | −0.74 ** | −0.60 ** | - | | | | | | | |
| 5. AGR | 0.09 * | −0.30 ** | −0.15 ** | −0.24 ** | −0.01 | - | | | | | | |
| 6. CONT | 0.33 ** | −0.32 ** | −0.17 ** | −0.28 ** | 0.05 | 0.16 * | - | | | | | |
| 7. ES | 0.17 ** | −0.75 ** | −0.45 ** | −0.64 ** | 0.33 ** | 0.14 ** | 0.26 ** | - | | | | |
| 8. OPN | −0.01 | −0.24 ** | −0.37 ** | −0.34 ** | 0.35 ** | 0.21 ** | 0.11 ** | 0.16 * | - | | | |
| 9. CMD | −0.13 ** | 0.56 ** | 0.45 ** | 0.57 ** | −0.31 ** | −0.13 ** | −0.19 ** | −0.51 ** | −0.51 ** | - | | |
| 10. ORG | 0.28 ** | −0.10 * | −0.02 | −0.08 | −0.04 | 0.09 * | 0.61 ** | 07 * | −0.15 ** | .06 | - | |
| 11. PS | 0.08 * | 0.19 ** | 0.13 ** | 0.19 ** | −0.06 | −0.11 ** | 0.18 ** | −0.13 ** | 0.07 * | 0.60 ** | 0.33 ** | - |
| 12. PEC | −0.13 ** | 0.23 ** | 0.20 ** | 0.25 ** | −0.09 * | −0.13 ** | −0.13 ** | −0.17 ** | −0.03 | 0.51 ** | −0.04 | 0.44 ** |

Note: MEQ: Morningness–Eveningness Questionnaire; NA: negative affect; SI: social inhibition; SI x NA; EXT: extraversion; AGR: agreeableness; CONT: conscientiousness; ES: emotional stability; OPN: openness to experiences; CMD: concerns over mistakes and doubts; ORG: organization; PS: personal standards; PEC: parental expectations and criticism. * Sig at <0.05, ** < 0.01.

Linear regression analysis (F (3,863) = 14.07, $p < 0.001$) determined that negative affect, but neither social inhibition nor the dimensional Type-D interaction, significantly predicted a diurnal preference (Step 1; 05% total variance explained). When including the Big Five personality traits of agreeableness, conscientiousness, and emotional stability (Step 2; 12% total variance explained), only negative affect and conscientiousness significantly predicted a diurnal preference (F (6,863) = 20.26, $p < 0.001$). Finally, after including the dimensions of multidimensional perfectionism (Step 3; 14% total variance explained), negative affect, conscientiousness, organization, personal standards, and parental perception remained the only significant predictors of the chronotype preference (F (10,863) = 14.69, $p < 0.001$; see Table 3). Here, increased levels of negative affect and parental evaluation and criticism appear to be related to a greater evening preference. In contrast, increased reports of conscientiousness and the perfectionism dimensions of organization and personal standards appear to be related to a greater morning preference.

**Table 3.** Linear regression analyses with chronotype preference as the dependent variable; significant personality correlates as predictors.

| Predictors | Adjusted R$^2$ | β | B | t | Sig. | BCa 95% CIs for B | |
|---|---|---|---|---|---|---|---|
| | | | | | | Lower | Upper |
| Step 1: | 0.05 | | | | | | |
| DS14: Negative Affect | | −0.26 | −0.36 | −3.62 | 0.001 ** | −0.54 | −0.17 |
| DS14: Social Inhibition | | −0.06 | −0.09 | −0.76 | 0.432 | −0.33 | 0.16 |
| DS14: SI x NA | | 0.09 | 0.01 | 0.77 | 0.436 | −0.01 | 0.02 |
| Step 2: | 0.12 | | | | | | |
| DS14: Negative Affect | | −0.19 | −0.26 | −2.42 | 0.012 * | −0.45 | −0.06 |
| DS14: Social Inhibition | | −0.10 | −0.15 | −1.32 | 0.170 | −0.37 | 0.09 |
| DS14: SI x NA | | 0.16 | 0.01 | 1.45 | 0.132 | 0.00 | 0.02 |
| TIPI: Agreeableness | | 0.01 | 0.12 | 0.41 | 0.707 | −0.48 | 0.78 |
| TIPI: Conscientiousness | | 0.29 | 2.13 | 8.62 | 0.001 ** | 1.60 | 2.66 |
| TIPI: Emotional Stability | | 0.01 | 0.08 | 0.24 | 0.821 | −0.50 | 0.66 |
| Step 3: | 0.14 | | | | | | |
| DS14: Negative Affect | | −0.18 | −0.25 | −2.34 | 0.014 * | −0.45 | −0.05 |
| DS14: Social Inhibition | | −0.08 | −0.12 | −1.07 | 0.276 | −0.33 | 0.11 |
| DS14: SI x NA | | 0.15 | 0.01 | 1.37 | 0.147 | 0.00 | 0.02 |
| TIPI: Agreeableness | | 0.02 | 0.14 | 0.48 | 0.660 | −0.44 | 0.80 |
| TIPI: Conscientiousness | | 0.18 | 1.30 | 4.12 | 0.001 ** | 0.63 | 1.93 |
| TIPI: Emotional Stability | | 0.01 | 0.04 | 0.14 | 0.891 | −0.56 | 0.67 |
| FMPS: Concerns and Doubts | | −0.06 | −0.05 | −1.09 | 0.281 | −0.16 | 0.06 |
| FMPS: Organization | | 0.13 | 0.24 | 3.02 | 0.001 ** | 0.09 | 0.38 |
| FMPS: Personal Standards | | 0.10 | 0.16 | 2.16 | 0.047 * | 0.00 | 0.30 |
| FMPS: Parental Expectation Criticism | | −0.09 | −0.13 | −2.45 | 0.020 * | −0.24 | −0.02 |

Note: Bootstrapped with 1000 bias-corrected resamples. MEQ: Morningness–Eveningness Questionnaire; DS14: Distressed Personality 14; TIPI: Ten-Item Personality Inventory; FMPS: Multidimensional Perfectionism Scale. * Sig at <0.05, ** < 0.01

## 4. Discussion

The present study aimed to determine the extent to which the dimensions of the five-factor model, Type-D personality, and multidimensional perfectionism are differentially associated with a diurnal preference in the general population. The results provide further evidence that specific aspects of the Big Five personality traits are associated with a diurnal preference. In line with the existing evidence, agreeableness, conscientiousness, and emotional stability were separately related to morningness [11–17]. More crucially, for the first time, we evidenced that the dimensions of Type-D personality and multidimensional perfectionism are associated with a diurnal preference. Here, negative affect, social inhibition, Type-D personality, and perfectionistic doubts and concerns, as well as an increased perception of critical parental evaluation, were independently related to eveningness. In contrast, organization and personal standards were related to a greater disposition for morningness. However, after accounting for the shared variance amongst the personality traits, only negative affect, conscientiousness, organization, personal standards, and parental perception were significantly associated with a diurnal preference.

Consistent with the literature to date, the current outcomes highlight greater reports of adaptive personality traits (i.e., conscientiousness, personal standards, organization) in relation to morningness [11–17]. Indeed, conscientiousness appears to be one of the most reliable predictors of a diurnal preference [29], whereby morning individuals have previously evidenced a more adaptive attitude towards future-oriented behavior, and greater reports of metacognitive behavior and impulse control, compared with their evening-type counterparts [15,30,31]. Supporting this notion, the current results evidenced that morningness is associated with the propensity to maintain a high standard of order, organization, and personal standards. In contrast, negative affect and perceived parental evaluation and criticism were the strongest predictors of eveningness in the present sample. When examining the individual role of Type-D personality, the regression analyses determined that negative affect explained 5% of the predictive variance in relation to diurnal preference. Following the addition of the TIPI, contentiousness added an additional 7%. Finally, the addition of the perfectionism dimensions added an additional 2%, where personal standards and parental perception were significantly associated with diurnal preference. These outcomes are in line with previous research highlighting the prevalence of potentially aversive traits amongst evening types, including emotional instability [14], risk-taking behavior [5], reduced behavioral activation and positive affect [6], trait-like anxiety [4], and psychopathy [7]. With this in mind, evening-type individuals often display difficulties in emotion regulation and adaptive coping, and particularly when faced with stress [29,30]. Likewise, emotional difficulties are frequently related to increased reactivity to negative emotions and poor self-concept (i.e., negative affect) [32], as well as dimensions of perfectionism [33,34], which are considered maladaptive (i.e., doubts and concerns over mistakes). Recent evidence indicates that evening-type individuals exhibit maladaptive metacognitive beliefs and emotion-regulation difficulties [30]. Here, morning-type individuals were more efficient at adequately deploying cognitive reappraisal strategies (i.e., the reinterpretation of an emotion-eliciting situation in a way that alters its meaning and changes its emotional impact) when required. Moreover, those with an evening disposition perceived the experience of worry as being negative and uncontrollable, showed distrust of their own memory, and selective attention towards their thoughts [30]. Cognitive processes of this nature may serve to accentuate the experiences of negative affect and perfectionism amongst evening-type individuals, whilst also contributing to the onset of psychiatric difficulties [30].

Optimal sleep occurs when the desired sleep time (based on the external 24 h clock time) is synchronized with an individual's internal circadian sleep timing [35]. Except for shiftwork, modern society largely revolves around a working schedule that favors morning types [3,36,37]. Indeed, this may explain the consistently evidenced relationships between morningness and beneficially adaptive personality traits [11–17]. Due to social and occupational factors, evening types must often initiate sleep outside of their circadian phases, which leads to difficulties in initiating and maintaining sleep, and increased daytime sleepi-

ness [38–41]. In the current context, the prolonged experience of circadian misalignment and the consequential symptoms may exacerbate the presentation of worry, rumination, and psychological distress amongst evening types. While this might be accentuated by certain personality traits (e.g., negative affect), the circadian rhythm (i.e., phase delay or advance) and subjective diurnal preference may disrupt the temporal stability of personality [42]. Indeed, individuals with delayed-sleep-phase disorder display significantly reduced levels of contentiousness and extraversion, alongside increased neuroticism [43]. More recently, young adults with delayed-sleep-phase disorder demonstrated deficits in psychosocial wellbeing, including social withdrawal, poor academic performance, and parental conflict [44].

Several limitations of the current study should be noted. The sample population was mostly female and predominantly students. Moreover, most of the sample was comprised of younger adults between 18 and 41 years (87%) who largely favored the evening (42%). Whilst younger adults typically prefer the evening [45], the outcomes may not be entirely generalizable to males, older adults, and the general population. Given the cross-sectional nature of the design, the outcomes remain limited in terms of the directionality and causality, as well as the potential vulnerability to an inflation bias between the variables. Moving forward, studies of a longitudinal design should clarify the extent to which personality may influence the diurnal preference using a more balanced sample in relation to age and sex. Finally, when examining the internal consistency of the current data, the TIPI subscales of agreeableness and openness to new experiences yielded significantly low values ($\alpha = 0.3$ for each). Although these subscales often display lower levels of internal consistency [25,46,47], the outcomes regarding agreeableness and openness to new experiences in the current study should be taken with caution.

In summary, the current outcomes offer further insight into the relationship between personality and diurnal preference. Here, we observed greater reports of adaptive personality traits in relation to morningness, whereas negative affect and perceived parental evaluation and criticism were related to eveningness. As the first study to examine the relationship between Type-D personality, multidimensional perfectionism, and diurnal preference, the current outcomes should be considered preliminary. Moving forward, the potential factors that underlie (e.g., emotion regulation, cognitive processes) the relationship between personality and diurnal preference should be explored.

**Author Contributions:** The study was designed and conceived by U.A. The data were collected by all the authors, and were analyzed and reported by J.C.S. The initial version of the manuscript was written by U.A. and J.C.S. Following this, input was sought from all the other authors. All authors have read and agreed to the published version of the manuscript.

**Funding:** This research received no external funding.

**Institutional Review Board Statement:** The study protocol was approved by the Sheffield Hallam University Research Ethics Committee (Ethical Approval ID: AMRKT/D&S-PSP-133, the date of approval 29 May 2017).

**Informed Consent Statement:** All participants provided informed consent online after indicating that they had carefully read and understood the study information provided, understood the terms and conditions of the study, and indicated that they indeed consented to taking part.

**Data Availability Statement:** The data will be made available upon reasonable request.

**Conflicts of Interest:** The authors declare no conflict of interest.

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
