# Peer review of "Diurnal Preference and Correlates of Multidimensional Perfectionism, Type-D Personality, and Big Five Personality Traits"

_2624-5175, doi:10.3390/clockssleep4030037_

Round 1

Reviewer 1 Report

The Authors looked for associations between diurnal preferences (morningness according to MEQ) and some personality traits ( DS14, TIPI and F-MPS - eleven scales in total). Methods and procedures are sufficiently described. Correlational and regression analyses were applied. The results proved greater trait-like punctuality related to morningness and showed some problems related to eveningness: negative affect and perceived parental evaluation and criticism.

Remarks:

Lines 60-67

Aim - the rationale for this correlational study is not clear/obvious. The statement that some of the relationships between diurnal preference and other personality traits ‘remain unexamined’ does not seem sufficient. Some hypotheses?

Line 110.

“Assessment of internal consistency yielded a Cronbach’s α of .65.” How did the authors calculate the internal consistency of a tool comprising five independent factors/scales? It makes little sense.

See Gosling et al., 2003: “Specifically, the Cronbach alphas were .68, .40, .50, .73, and .45 for the Extraversion, Agreeableness, Conscientiousness, Emotional Stability, and Openness to Experience scales respectively.”

Lines 121-122:  Something strange happened with this sentence…

Line 128: “Each personality type were entered as separate predictors in the model.”  Is that correct? Maybe you meant ‘trait’?  

Table 1, title: TII -> TIPI

Table 2.: Why is OPN bolded?

I wonder whether including age as a factor would bring some new interpretation. Assuming that evening preference is linked to younger age and parental criticism regards mainly young people, it would make more complete picture of the relationships described.

Besides, the population investigated does not seem to be ‘the general population’, considering the relatively low mean age (22,9 years; distribution not given, but hardly expected to be normal).

Overall, the Discussion section offers little explanation as to the possible mechanisms of the observed relationships/correlations, except of inadequate sleep timing in late chronotypes. I would appreciate, for example, some considerations on ‘morning-oriented society’ in which we all function, and the general positive attitude towards ‘early birds’ ;) 

Author Response

Response to reviewer one

The Authors looked for associations between diurnal preferences (morningness according to MEQ) and some personality traits ( DS14, TIPI and F-MPS - eleven scales in total). Methods and procedures are sufficiently described. Correlational and regression analyses were applied. The results proved greater trait-like punctuality related to morningness and showed some problems related to eveningness: negative affect and perceived parental evaluation and criticism.

Comment: Lines 60-67: Aim - the rationale for this correlational study is not clear/obvious. The statement that some of the relationships between diurnal preference and other personality traits ‘remain unexamined’ does not seem sufficient. Some hypotheses? 

Response: The main rationale was that this is the first study to examine type-d and perfectionism in the current context. In line with your suggestion, we have added hypotheses.

Specifically, we write (changes in bold): “To the best of our knowledge, possible relationship(s) between diurnal preference and dimensions of multidimensional perfectionism (excessive personal standards and overly critical self-evaluation) and Type-D (joint experience of negative affect and social inhibition) personality remain unexamined. Therefore, this exploratory study aimed to determine the extent to which: (i) dimensions of five-factor model, Type-D personality, and multidimensional perfectionism are directionally associated with diurnal preference in the general population; and (ii) any empirically supported relationships remained after accounting for shared variance between traits. Considering previous outcomes [4-17], it was hypothesized that eveningness to be related to negatively perceived aspects of personality. In contrast, we expected the preference for morningness to be associated with more positively oriented traits. The present study is the first to examine diurnal preference in relation to Type-D personality and multidimensional perfectionism. However, given mixed evidence concerning which dimensions of the big-five personality traits are related to chronotype, no a-priori hypotheses are made in relation to this question.

Comment: Line 110. “Assessment of internal consistency yielded a Cronbach’s α of .65.” How did the authors calculate the internal consistency of a tool comprising five independent factors/scales? It makes little sense. 

See Gosling et al., 2003: “Specifically, the Cronbach alphas were .68, .40, .50, .73, and .45 for the Extraversion, Agreeableness, Conscientiousness, Emotional Stability, and Openness to Experience scales respectively.” 

Response: These have now been calculated accordingly.

Specifically, we write (changes in bold): “Assessment of internal consistency yielded a Cronbach’s α of .74, .30, .50, 70 and .30 for the extraversion, agreeableness, conscientiousness, emotional stability and openness to new experiences subscales respectively.

Comment: Lines 121-122:  Something strange happened with this sentence… 

Line 128: “Each personality type were entered as separate predictors in the model.”  Is that correct? Maybe you meant ‘trait’?  

Response: Thank you for highlighting these errors. We now write: Correlational analyses (Pearson’s bivariate) examined the relationship between each personality trait and chronotype preference. Next, a bootstrapped hierarchical linear regression analysis was used to assess the predictive value of significant correlational associations between chronotype and the assessed personality characteristics. Bootstrapping is a robust alternative to standard parametric estimates, when the assumptions around the latter may be violated [24]. Bootstrapping with 1000 bias-corrected and accelerated (BCa) resamples and 95% confidence intervals were used, as this analytic approach allows for a more robust estimation of the regression coefficients [25]. Each personality trait was entered as separate predictors in the model.”

Comment: Table 1, title: TII -> TIPI 

Response: Thank you for highlighting this error, which has now been corrected. As a side note, it seems that someone has pre copy edited the manuscript which has led to some major formatting inconsistencies, which I have also addressed.

Comment: Table 2.: Why is OPN bolded? 

Response: I think this was a result of the reformatting as mentioned above. This has now been corrected, and the tables are back to being consistent – unless it is edited again before review – hopefully not!

Comment: I wonder whether including age as a factor would bring some new interpretation. Assuming that evening preference is linked to younger age and parental criticism regards mainly young people, it would make more complete picture of the relationships described. Besides, the population investigated does not seem to be ‘the general population’, considering the relatively low mean age (22,9 years; distribution not given, but hardly expected to be normal). 

Response: After looking at the distribution (histogram), the picture falls in like with the standard deviation yielded (mean age = 22.93 ± 9.67), with only 13% of the sample aged above 25 years. Indeed, whilst diurnal preference remains relatively stable, we generally observe a circadian phase advance in older adulthood. As age was not the current focus, and this work is a short communication, we have provided some comment in the discussion. I think additional research with a more balance sample is required moving forward.

Specifically, we write (changes in bold): Several limitations of the current study should be noted. The sample population was mostly female and predominantly students. Moreover, most of the sample comprised of younger adults between 18-14 years (87%). Therefore, the outcomes may not be entirely generalizable to males, older adults, and the general population. Given the cross-sectional nature of the design, the outcomes remain limited in terms of directionality and causality, and potential vulnerability to an inflation bias between variables. Therefore, studies of a longitudinal design should clarify the extent to which personality may influence diurnal preference using a more balanced sample in relation to age and sex.

Comment: Overall, the Discussion section offers little explanation as to the possible mechanisms of the observed relationships/correlations, except of inadequate sleep timing in late chronotypes. I would appreciate, for example, some considerations on ‘morning-oriented society’ in which we all function, and the general positive attitude towards ‘early birds’ ;) 

Response: In response to the second reviewer, we have now referred to this bias for morning types. As an extreme evening type with idiopathic hypersomnia, I find the current morning orientated society quite frustrating!

Specifically, we write (changes in bold): “Optimal sleep occurs when desired sleep time (based on external 24hr clock time) is synchronized with an individual’s internal circadian sleep timing [34]. Indeed, this may explain the consistently evidenced relationships between morningness and beneficially adaptive personality traits [11-17]. Except for shiftwork, modern society largely revolves around a working schedule favoring morning-types [35-38]. Due to social and occupational factors, evening-types must often initiate sleep outside of their circadian phase, leading to difficulties in initiating and maintaining sleep and increased daytime sleepiness [39-41]. In the current context, the prolonged experience of circadian misalignment and consequential symptoms may exacerbate the presentation of worry, rumination, and psychological distress amongst evening-types. While this might be accentuated by certain personality traits (e.g., negative affect), the circadian rhythm (i.e., phase delay or advance) and subjective diurnal preference may disrupt the temporal stability of personality [42]. Indeed, individuals with delayed-sleep phase disorder display significantly reduced levels of contentiousness and extraversion alongside increased neuroticism [43]. More recently, young adults with delayed-sleep phase disorder demonstrated deficits in psychosocial wellbeing including social withdrawal, academic performance, and parental conflict [44].

Reviewer 2 Report

This paper addresses the worthwhile topic of the relations between morningness-eveningness and a variety of personality variables in a large sample of general population adults (N = 864). It is well-written and the statistics are generally appropriate and well reported – although see some caveats below. As noted in the Discussion, it is highly beneficial to examine all the personality variables together. But there are some issues with interpretation and wording.

1.     Bivariate correlations are described as one variable being “independently related to” another variable. This terminology is a little misleading because “independent” has a statistical meaning implying “independent of other variables that are controlled for”, which is not what the authors mean. It would be better to say “separately” (or some other less ambiguous term.)

2.     It was appropriate to analyze the MPQ as a continuous bipolar variable. But, in order to understand the meaning of this variable in the current sample, we need to know more about its distribution. It would be useful to state (a) the theoretical midpoint of the scale (i.e., no preference for either morningness or eveningess) and (b) the modality and symmetry/skew of the distribution of scores (or perhaps the % of cases above and below the theoretical midpoint). In an extreme example (clearly not the case here), the authors could report an association with eveningness whereas all the participants actually prefer morningness to one extent or another.

3.     Line 158-160: “…are directionally associated with diurnal preference in the general population.” If this means a causal direction it is incorrect since the correlational results do not indicate causality (as noted in the Discussion). If it means diurnal direction, then it should be “are associated with diurnal preference direction in the general population.”

4.     Overall, the authors do a good job of avoiding causal wording in describing results. However, the possibilities of diurnal preference influencing personality and personality influencing diurnal preference are relevant to interpreting and understanding the results. It would be helpful if the authors could be more explicit (briefly) about when they are considering each – I know there is no definitive answer and it is fine to be a little speculative. Also note:

(a)   Lines 35-37. “Personality may influence the directional nature of one’s diurnal preference. Indeed, various personality traits are differentially associated with a greater preference for the morning or evening [4-17].” The first sentence is causal (OK since theoretical) but evidence is not causal.

(b)   Lines 199 – 205 consider a chronotype => personality possibility.

5.     Why is there a single Cronbach’s alpha for the TIPI scale of the 5 Big Five personality traits? The total score across the 5 traits is meaningless. (In contrast, the authors correctly report an alpha for each of the subscales of the F-MPS.)

6.     Line 122 typo.

7.     No rationale is provided for including the SI x NA interaction.

8.     Although the stepwise regression procedure is clearly described in the Statistical Analyses section, no rationale is provided for conducting a stepwise multiple regression rather than a simultaneous regression. The final step 3 is what is interpreted and is essentially a simultaneous regression of all variables. The percent variance explained at each step is reported, but never interpreted or commented on. Wouldn’t it be simpler just to do a single simultaneous regression?

9.     Although an important endeavor, I find problems with the way the authors attempt to find a conceptual connection between diurnal preference and personality variables (also see my comment on causality). Although possible correlates, conscientiousness, personal standards, and organization (line 173) do not constitute “trait-like punctuality”. Citations 11-14 are used to back up this argument, but the only hit in a PsycInfo search of “punctualilty” and the first authors was a different paper by Randler, first authored by Werner.

Similarly for conscientiousness and "scheduled goal-directed behaviour"; they are simply not the same construct. The term “scheduled” does appear in the Hogben article cited [15]. Is the citation 24 correct? It seems to be a statistics paper. Perhaps the authors could revise the wording of these ideas in a way that makes better sense.

Author Response

Response to reviewer two

This paper addresses the worthwhile topic of the relations between morningness-eveningness and a variety of personality variables in a large sample of general population adults (N = 864). It is well-written and the statistics are generally appropriate and well reported – although see some caveats below. As noted in the Discussion, it is highly beneficial to examine all the personality variables together. But there are some issues with interpretation and wording.

Comment: Bivariate correlations are described as one variable being “independently related to” another variable. This terminology is a little misleading because “independent” has a statistical meaning implying “independent of other variables that are controlled for”, which is not what the authors mean. It would be better to say “separately” (or some other less ambiguous term.)

Response: This makes sense, we chose to use independent in the sense that the correlation is just the relationship between two variables. For greater clarity, we have now rephrased to: “separately”.

Comment: It was appropriate to analyze the MPQ as a continuous bipolar variable. But, in order to understand the meaning of this variable in the current sample, we need to know more about its distribution. It would be useful to state (a) the theoretical midpoint of the scale (i.e., no preference for either morningness or eveningess) and (b) the modality and symmetry/skew of the distribution of scores (or perhaps the % of cases above and below the theoretical midpoint). In an extreme example (clearly not the case here), the authors could report an association with eveningness whereas all the participants actually prefer morningness to one extent or another.

Response: We have now visually presented the distribution of MEQ scores using a histogram. As depicted, the distribution appears to be relatively normal.

Figure 1. Distribution of total MEQ scores in the current sample, where scores between: 16-41 indicate an evening preference; 59-86 indicate a morning preference. 

Comment: Line 158-160: “…are directionally associated with diurnal preference in the general population.” If this means a causal direction it is incorrect since the correlational results do not indicate causality (as noted in the Discussion). If it means diurnal direction, then it should be “are associated with diurnal preference direction in the general population.”

Response: This was a major typographical error. Here, we meant to write: differentially. This word has now been replaced.

Comment: Overall, the authors do a good job of avoiding causal wording in describing results. However, the possibilities of diurnal preference influencing personality and personality influencing diurnal preference are relevant to interpreting and understanding the results. It would be helpful if the authors could be more explicit (briefly) about when they are considering each – I know there is no definitive answer and it is fine to be a little speculative. Also note:

Response: We have added more discussion that may tease apart the positive vs negative aspects, and why this might be in the discussion.

Specifically, we write (changes in bold): “Consistent with the literature to date, the current outcomes highlight greater reports of adaptive personality traits (i.e., conscientiousness, personal standards, organization) in relation to morningness [11-17]. Indeed, conscientiousness appears to be one of the most reliable predictors of diurnal preference [29] whereby morning individuals are previously evidenced to exhibit systematic and scheduled goal-directed behavior and impulse control compared to their evening-type counterparts [15, 29]. Supporting this notion, the current results evidenced morningness to be associated with the propensity to maintain a high standard of order, organization, and personal standards. In contrast, negative affect and perceived parental evaluation and criticism were the strongest predictors of eveningness in the present sample. When examining the individual role of Type-d personality, regression analyses determined that negative affect explained 5% of predictive variance in relation to diurnal preference. Following addition of the TIPI, contentiousness added an additional 7%. Finally, the addition of perfectionism dimensions added an additional 2%, where personal standards, and parental perception were significantly associated with diurnal preference. These outcomes are in line with previous research highlighting the prevalence of potentially aversive traits amongst evening-types, including emotional instability [14], risk-taking behavior [5], reduced behavioral activation and positive affect [6], trait-like anxiety [4], and psychopathy [7]. With that in mind, evening-type individuals often display difficulties in emotion regulation and adaptive coping, particularly when faced with stress [29-30]. Likewise, emotional difficulties are frequently related to increased reactivity to negative emotions and poor self-concept (i.e., negative affect) [31] and dimensions of perfectionism [32-33] which are considered maladaptive (i.e., doubts and concerns over mistakes). Recent evidence indicates that evening-type individuals exhibit maladaptive metacognitive beliefs and emotion regulation difficulties [30]. Here, morning-type individuals were more efficient in adequately deploying cognitive reappraisal strategies (i.e., reinterpretation of an emotion-eliciting situation in a way that alters its meaning and changes its emotional impact) when required. Moreover, those with an evening disposition perceived the experience of worry as being negative and uncontrollable, showed distrust of their own memory, and selective attention towards their thoughts [30]. Cognitive processes of this nature may serve to accentuate the experience of negative affect and perfectionism amongst evening-type individuals, whilst also contributing to the onset of psychiatric difficulties [30].

Optimal sleep occurs when desired sleep time (based on external 24hr clock time) is synchronized with an individual’s internal circadian sleep timing [34]. Except for shiftwork, modern society largely revolves around a working schedule favoring morning-types [35-38]. Indeed, this may explain the consistently evidenced relationships between morningness and beneficially adaptive personality traits [11-17]. Due to social and occupational factors, evening-types must often initiate sleep outside of their circadian phase, leading to difficulties in initiating and maintaining sleep and increased daytime sleepiness [39-41]. In the current context, the prolonged experience of circadian misalignment and consequential symptoms may exacerbate the presentation of worry, rumination, and psychological distress amongst evening-types. While this might be accentuated by certain personality traits (e.g., negative affect), the circadian rhythm (i.e., phase delay or advance) and subjective diurnal preference may disrupt the temporal stability of personality [42]. Indeed, individuals with delayed-sleep phase disorder display significantly reduced levels of contentiousness and extraversion alongside increased neuroticism [43]. More recently, young adults with delayed-sleep phase disorder demonstrated deficits in psychosocial wellbeing including social withdrawal, academic performance, and parental conflict [44].

Comment: Lines 35-37. “Personality may influence the directional nature of one’s diurnal preference. Indeed, various personality traits are differentially associated with a greater preference for the morning or evening [4-17].” The first sentence is causal (OK since theoretical) but evidence is not causal. 

Response: This has now been rephrased.

Specifically, we write (changes in bold): Personality may theoretically influence the directional nature of one’s diurnal preference. Indeed, various personality traits are differentially associated with a greater preference for the morning or evening [4-17].”

Comment: Lines 199 – 205 consider a chronotype => personality possibility.

Response: We have added the following.

Specifically, we write (changes in bold): “Optimal sleep occurs when desired sleep time (based on external 24hr clock time) is synchronized with an individual’s internal circadian sleep timing [34]. Indeed, this may explain the consistently evidenced relationships between morningness and beneficially adaptive personality traits [11-17]. Except for shiftwork, modern society largely revolves around a working schedule favoring morning-types [35-38]. Due to social and occupational factors, evening-types must often initiate sleep outside of their circadian phase, leading to difficulties in initiating and maintaining sleep and increased daytime sleepiness [39-41]. In the current context, the prolonged experience of circadian misalignment and consequential symptoms may exacerbate the presentation of worry, rumination, and psychological distress amongst evening-types. While this might be accentuated by certain personality traits (e.g., negative affect), the circadian rhythm (i.e., phase delay or advance) and subjective diurnal preference may disrupt the temporal stability of personality [42]. Indeed, individuals with delayed-sleep phase disorder display significantly reduced levels of contentiousness and extraversion alongside increased neuroticism [43]. More recently, young adults with delayed-sleep phase disorder demonstrated deficits in psychosocial wellbeing including social withdrawal, academic performance, and parental conflict [44].

Comment: Why is there a single Cronbach’s alpha for the TIPI scale of the 5 Big Five personality traits? The total score across the 5 traits is meaningless. (In contrast, the authors correctly report an alpha for each of the subscales of the F-MPS.)

Response: These have now been calculated accordingly.

Specifically, we write (changes in bold): “Assessment of internal consistency yielded a Cronbach’s α of .74, .30, .50, 70 and .30 for the extraversion, agreeableness, conscientiousness, emotional stability and openness to new experiences subscales respectively.

Comment: Line 122 typo.

Response: Thank you for highlighting this error. We now write: Correlational analyses (Pearson’s bivariate) examined the relationship between each personality trait and chronotype preference.”

Comment: No rationale is provided for including the SI x NA interaction.

Response: Type D personality was traditionally conceptualized as a categorical variable with individuals scoring above a threshold on both SI and NA are classified as Type D (Denollet, 2005), however recent recommendations propose that it is alternatively represented as a dimensional construct (Stevenson & Williams, 2014; Ferguson et al., 2009). Therefore, in line with previous studies Type D was considered as both a categorical and a continuous variable within the current study.

We have added a sentence in the measures section for greater clarity. Specifically, we write (changes in bold): “To analyse type-d as a dimensional construct, a continuous measure of type-d was computed using the arithmetic product of SI and NA scores. This is in line with recent studies examining the dimensional type-d construct [21-25].”

Comment: Although the stepwise regression procedure is clearly described in the Statistical Analyses section, no rationale is provided for conducting a stepwise multiple regression rather than a simultaneous regression. The final step 3 is what is interpreted and is essentially a simultaneous regression of all variables. The percent variance explained at each step is reported, but never interpreted or commented on. Wouldn’t it be simpler just to do a single simultaneous regression?

Response: We choose to input each personality measure in separate steps to examine the percentage of variance related to each trait type. Whilst reported in the table and text, we have now provided further comment in the discussion.

Specifically, we write (changes in bold): “In contrast, negative affect and perceived parental evaluation and criticism were the strongest predictors of eveningness in the present sample. When examining the individual role of Type-d personality, regression analyses determined that negative affect explained 5% of predictive variance in relation to diurnal preference. Following addition of the TIPI, contentiousness added an additional 7%. Finally, the addition of perfectionism dimensions added an additional 2%, where personal standards, and parental perception were significantly associated with diurnal preference. These outcomes are in line with previous research highlighting the prevalence of potentially aversive traits amongst evening-types, including emotional instability [14], risk-taking behavior [5], reduced behavioral activation and positive affect [6], trait-like anxiety [4], and psychopathy [7]. With that in mind, evening-type individuals often display difficulties in emotion regulation and adaptive coping, particularly when faced with stress [29-30]. Likewise, emotional difficulties are frequently related to increased reactivity to negative emotions and poor self-concept (i.e., negative affect) [31] and dimensions of perfectionism [32-33] which are considered maladaptive (i.e., doubts and concerns over mistakes). Recent evidence indicates that evening-type individuals exhibit maladaptive metacognitive beliefs and emotion regulation difficulties [30]. Here, morning-type individuals were more efficient in adequately deploying cognitive reappraisal strategies (i.e., reinterpretation of an emotion-eliciting situation in a way that alters its meaning and changes its emotional impact) when required. Moreover, those with an evening disposition perceived the experience of worry as being negative and uncontrollable, showed distrust of their own memory, and selective attention towards their thoughts [30]. Cognitive processes of this nature may serve to accentuate the experience of negative affect and perfectionism amongst evening-type individuals, whilst also contributing to the onset of psychiatric difficulties [30].”

Comment: Although an important endeavor, I find problems with the way the authors attempt to find a conceptual connection between diurnal preference and personality variables (also see my comment on causality). Although possible correlates, conscientiousness, personal standards, and organization (line 173) do not constitute “trait-like punctuality”. Citations 11-14 are used to back up this argument, but the only hit in a PsycInfo search of “punctualilty” and the first authors was a different paper by Randler, first authored by Werner.

Response: This has now been rephrased throughout, including the abstract.

In the discussion, we write (changes in bold): Consistent with the literature to date, the current outcomes highlight greater reports of adaptive personality traits (i.e., conscientiousness, personal standards, organization) in relation to morningness [11-17]. Indeed, conscientiousness appears to be one of the most reliable predictors of diurnal preference [29] whereby morning individuals are previously evidenced to exhibit systematic and scheduled goal-directed behavior and impulse control compared to their evening-type counterparts [15, 29].”

Comment: Similarly for conscientiousness and "scheduled goal-directed behaviour"; they are simply not the same construct. The term “scheduled” does appear in the Hogben article cited [15]. Is the citation 24 correct? It seems to be a statistics paper. Perhaps the authors could revise the wording of these ideas in a way that makes better sense.

Response: This was based on amended order of references, 15 and 29. The submission was not in numerical format as was copyedited (I think) prior to review. This has thrown the formatting consistency off somewhat, which we have corrected.

  • Hogben, A.L., Ellis, J., Archer, S.N. and von Schantz, M., 2007. Conscientiousness is a predictor of diurnal preference. Chronobiology International24(6), pp.1249-1254.
  • Taylor BJ, Bowman MA, Brindle A, Hasler BP, Roecklein KA, Krafty RT, Matthews KA, Hall MH. Evening chronotype, alcohol use disorder severity, and emotion regulation in college students. Chronobiology International. 2020 Dec 1;37(12):1725-35.

Round 2

Reviewer 2 Report

The authors have satisfactorily addressed many of the points I raised in my original review. However, a few issues have not been addressed, even though there has been a reply. The numbers that follow are those from my original review.

 2. In the revision, the histogram of the MEQ scores is now provided; however, there is no text that indicates why it is there or how it is relevant. Also, there is also no mention in the Measures section (as requested) of what MEQ value represents the theoretical midpoint so that values above it represent a tendency to morningness and values below it represent a tendency to eveningness. The authors should note either (a) what proportions  of the sample fall above and below this midpoint qualifying as morning-ish and evening-ish (based on the histogram, around 78% of the sample is in the more evening range, below 51), or (b) the proportions in the morning, evening and intermediate ranges as defined in the Horne paper, or (c) both. Doing this does not negate treating chronotype preference as a dimensional variable. It is bipolar and not just more or less of a single construct. Note that this comment is about addressing an issue, not just rote inclusion of more information.

 5. Cronbach alphas for each of the big five personality measures are now appropriately provided. However, three of them are quite low (.5, .3, .3) raising questions about the adequacy of these measures. At minimum this needs to be noted in the limitations section.

 7. My original comment was about the absence of a rationale for including in the regression the SI x NA interaction. The authors’ response was about treating the variables dimensionally, which is not a problem and is not the issue I raised.

 10.     Line 194-95. The reference numbers have been corrected. But I still don’t see any mention in those articles of “scheduled goal-directed behavior”. It is not clear how this idea is supported in the literature or in this study.

Author Response

Response to reviewer two

Comment: In the revision, the histogram of the MEQ scores is now provided; however, there is no text that indicates why it is there or how it is relevant. Also, there is also no mention in the Measures section (as requested) of what MEQ value represents the theoretical midpoint so that values above it represent a tendency to morningness and values below it represent a tendency to eveningness. The authors should note either (a) what proportions  of the sample fall above and below this midpoint qualifying as morning-ish and evening-ish (based on the histogram, around 78% of the sample is in the more evening range, below 51), or (b) the proportions in the morning, evening and intermediate ranges as defined in the Horne paper, or (c) both. Doing this does not negate treating chronotype preference as a dimensional variable. It is bipolar and not just more or less of a single construct. Note that this comment is about addressing an issue, not just rote inclusion of more information.

Response: We have now included categorical scoring information in the measure section, and the percentage information in the results. Indeed, there is a greater overall preference for the evening in this sample, likely due to the sample populations age group. This is also highlighted in the discussion.

Specifically, we write (changes in bold): “Chronotype preference was determined using the 19-item Morningness–Eveningness Questionnaire (MEQ: 19), which asks about individuals sleep timing and schedules (e.g., “If you got into bed at 11 PM, how tired would you be?’ (0=not at all tired; 5=very tired) and ‘‘During the first half hour after you wake up in the morning, how do you feel?” (1=very tired; 4=very refreshed). Total scores range between 16-86, and higher scores indicate a greater preference for morningness. In contrast, lower scores indicate a disposition for eveningness. Total scores between 16-41, 42-58 and 59-86 indicate a preference for morningness, neither (intermediate) or eveningness respectively. Items are summed to form an index of chronotype. A good level of internal consistency was yielded (Cronbach’s α of .86).”

And…

Mean scores for chronotype preference and each personality trait are presented in Table 1. Normality of the MEQ data was assessed using a histogram indicating the distribution to be skewed in favor of eveningness (see Figure 1). More specifically, 7.1% of the sample indicated a morning preference, 51.2% an intermediate preference, and 41.8% an evening preference.

And…

“Several limitations of the current study should be noted. The sample population was mostly female and predominantly students. Moreover, most of the sample comprised of younger adults between 18-14 years (87%) who largely favored the evening (42%). Whilst younger adults typically prefer the evening [45],the outcomes may not be entirely generalizable to males, older adults, and the general population. Given the cross-sectional nature of the design, the outcomes remain limited in terms of directionality and causality, and potential vulnerability to an inflation bias between variables. Therefore, studies of a longitudinal design should clarify the extent to which personality may influence diurnal preference using a more balanced sample in relation to age and sex. Finally, when examining the internal consistency of the current data, the TIPI subscales of agreeableness and openness to new experiences yielded significantly low values (α = .3 respectively). Although these subscales often display lower levels of internal consistency [25, 46-47], the outcomes regarding agreeableness and openness to new experiences in the current study should be taken with caution.

Comment: Cronbach alphas for each of the big five personality measures are now appropriately provided. However, three of them are quite low (.5, .3, .3) raising questions about the adequacy of these measures. At minimum this needs to be noted in the limitations section.

Response: Indeed, this appears to be quite common with this measure. For example, See Gosling et al., 2003: “Specifically, the Cronbach alphas were .68, .40, .50, .73, and .45 for the Extraversion, Agreeableness, Conscientiousness, Emotional Stability, and Openness to Experience scales respectively.”

This is now highlighted in the limitations section, as can be seen in response to the comment above.

Comment: My original comment was about the absence of a rationale for including in the regression the SI x NA interaction. The authors’ response was about treating the variables dimensionally, which is not a problem and is not the issue I raised.

Response: It’s unclear why a rationale would be required when considering that the interaction was significantly related to diurnal preference and most personality traits. This should be clear to most readers with statistical knowledge.

Comment: Line 194-95. The reference numbers have been corrected. But I still don’t see any mention in those articles of “scheduled goal-directed behavior”. It is not clear how this idea is supported in the literature or in this study.

Response: The wrong reference was numbered. 29 should have been 30. Whilst the authors do not directly use the same term, the metacognitive beliefs (i.e., being more assertive, confident, self-aware, in control) would fall in line with being considered goal-directed behaviour. This has now been rephrased, with an additional supporting reference.

Specifically, we write (changes in bold): “Indeed, conscientiousness appears to be one of the most reliable predictors of diurnal preference [29] whereby morning individuals are previously evidenced to exhibit a more adaptive attitude towards future-oriented behavior, and greater reports of metacognitive behaviour and impulse control compared to their evening-type counterparts [15, 30, 48].